# Transparent and Robust RAG: Adaptive-Reward Reinforcement Learning for Decision Traceability

## Abstract

Retrieval-Augmented Generation (RAG) delivers substantial value in knowledge-intensive applications. Many recent works use reinforcement learning (RL) to effectively elicit strong reasoning in *RAG generators*. However, two key challenges remain unresolved: **(1) Transparency**: most prior methods do not explicitly indicate which references are actually used during the reasoning that leads to the final answer, limiting interpretability and visibility; **(2) Stability**: the KL divergence estimator used in existing RL-based approaches may cause gradient spikes, which can lead to unstable training. To address these challenges, we propose **A**daptive-**R**ewarded **E**vidence **N**avigation **A**gent (**ARENA**), a transparent and robust RAG generator framework trained via RL with designed rewards. Based on our proposed structured protocol, KL divergence stabilization, and adaptive reward calculation modules, **ARENA** enables RAG generator to identify key evidence, perform structured reasoning, and generate answers with interpretable decision traces. Applied to Qwen2.5-7B-Instruct and Llama3.1-8B-Instruct, abundant experiments with various baselines demonstrate that our model achieves highly transparent outputs with 10–30% accuracy improvements across three multi-hop QA datasets, which is comparable with advanced closed-source LLMs (e.g., OpenAI-o1, DeepSeek-R1). Further analyses show ARENA has strong generalization to unseen datasets and tasks. Our models and codes will be publicly released[1].

## 1 Introduction

Retrieval-Augmented Generation (RAG) has become a powerful paradigm for enhancing large language models (LLMs) with non-parametric knowledge (Gao et al., 2023; Gupta et al., 2024; Li et al., 2025b; Zhou et al., 2025), particularly in knowledge-intensive domains such as medicine (Li et al., 2024a), law (Wiratunga et al., 2024), and finance (Setty et al., 2024). Beyond accuracy gains from retrieved context (Lewis et al., 2020; Yu, 2022), practical deployment further requires transparency, auditability, and decision traceability, so that one can verify which evidence supports which reasoning steps and the final answer (Singh et al., 2024; Wei Jie et al., 2024; Li et al., 2024b).

Despite rapid progress, important gaps remain. Recent RL-based work emphasizes active acquisition of information, interleaving search and reasoning when initial documents are insufficient (Gao et al., 2025; Li et al., 2025a). However, two major challenges are still unresolved. **Transparency:** open-source systems generally do not make explicit or verifiable which retrieved references are actually used during the reasoning that leads to the final answer (Song et al., 2025a; Chen et al., 2025b), while strong closed-source commercial models provide citation without disclosing training or data acquisition strategies (Team, 2025; OpenAI, 2025); moreover, the literature lacks a quantitative assessment of citation relevance accuracy (Jia et al., 2024; Yu et al., 2023; Wei et al., 2024). **Stability:** current RL-based methods for RAG largely adopt GRPO (Guo et al., 2025), which shown to elicit powerful reasoning in math (Zeng et al., 2025) and code (Liu et al., 2024), but the KL divergence estimator used in these methods may induce sharp gradient spikes during training, and there is little theoretical or empirical analysis of how the choice of estimator affects training stability and generation quality in RAG (Tang & Munos, 2025; Yu et al., 2025).

---

[1]Our code has been submitted as a supplementary ZIP and will be released on GitHub upon paper acceptance.

To address these challenges, we propose **A**daptive-**R**ewarded **E**vidence **N**avigation **A**gent (**ARENA**), a transparent and robust RAG generator trained with reinforcement learning. First, ARENA enforces a structured output protocol that explicitly includes the selected references, step-wise reasoning traces, and the final answer, rendering the entire evidence–reasoning–answer chain **auditable**. Second, it selects a more numerically **stable** KL divergence estimator to mitigate gradient spikes during optimization. Third, it employs **adaptive, task-specific** reward computation aligned with multi-hop QA (Yang et al., 2018; Ho et al., 2020; Trivedi et al., 2022), encouraging correct answers, faithful citation usage, and well-formed decision traces.Applied to open-source backbones such as Qwen2.5-7B-Instruct (Yang et al., 2024) and Llama3.1-8B-Instruct (Grattafiori et al., 2024), ARENA produces highly transparent outputs and delivers 10–30% accuracy improvements across three datasets while approaching advanced closed-source LLMs. Moreover, our generalization experiments demonstrate that ARENA transfers well across different tasks, backbones, and retrieval settings, our ablation studies further verify the effectiveness of each individual component.

Our main contributions are listed as follows:

- We propose **ARENA**, a reinforcement learning framework for RAG generator that improves both transparency and robustness. It integrates structured output protocol for explicit evidence–reasoning–answer traceability, KL-divergence stabilization that mitigates gradient spikes, and adaptive task-specific rewards tailored for multi-hop QA.

- We highlight the choice of KL divergence estimator will impact the stability and performance of RL in RAG, with both theoretical and empirical analysis.

- We validate our approach on three multi-hop QA benchmarks (HotpotQA, 2Wiki, and MuSiQue), where ARENA delivers 10–30% improvements in EM, F1, and LLM-as-a-judge metrics while maintaining superior transparency, approaching the performance of closed-source commercial models and demonstrating the effectiveness of our framework.

## 2 RELATED WORK

### 2.1 RAG FOUNDATIONS AND THE TRANSPARENCY GAP

RAG grounds model outputs in external evidence (Guu et al., 2020; Gao et al., 2023; Lewis et al., 2020), improving factual consistency on knowledge-intensive tasks such as open-domain QA (Shuster et al., 2021; Chen et al., 2024). Complex multi-hop QA tasks (Yang et al., 2018; Ho et al., 2020; Trivedi et al., 2022; Press et al., 2023) are particularly challenging for RAG, where irrelevant or noisy context can mislead generation and cause factual errors (Petroni et al., 2020; Creswell et al., 2023). To mitigate such risks and improve interpretability, prior work explores citation-based auditing and related techniques (Huang & Chang, 2024; Jia et al., 2024; Yu et al., 2023; Wei et al., 2024). However, despite these efforts, existing studies still lack explicit and readily evaluable citation labels that make clear which retrieved references support intermediate reasoning and the final answer.

### 2.2 SEARCH-CENTRIC RAG: PROGRESS AND REMAINING STABILITY ISSUES

Strengthening retrieval has been shown to improve RAG performance, and its benefits tend to grow with model scale (Mallen et al., 2023; Shi et al., 2024). Early approaches rely on prompt-based techniques to enhance retrieval (Press et al., 2023; Ma et al., 2023; Trivedi et al., 2023; Kim et al., 2024). More recently, research has focused on settings where initially retrieved documents are insufficient to answer the query, prompting models to actively acquire additional information until they can infer the answer. Supervised finetuning approaches synthesize retrieval trajectories and apply multi-criteria filtering to construct high-quality training data (Wang et al., 2025b; Sun et al., 2025b), while reinforcement learning optimizes search policies with reward feedback through online real search (Song et al., 2025b; Chen et al., 2025b) or offline simulated search (Mei et al., 2025; Sun et al., 2025a). Methodologically, most RL-based methods adopt GRPO or related policy-gradient variants (Shao et al., 2024; Hu, 2025); prior studies further suggest that commonly used KL divergence estimators can make training unstable (Tang & Munos, 2025; Yu et al., 2025). This observation motivates us to explore more stable KL divergence estimators for RAG training.

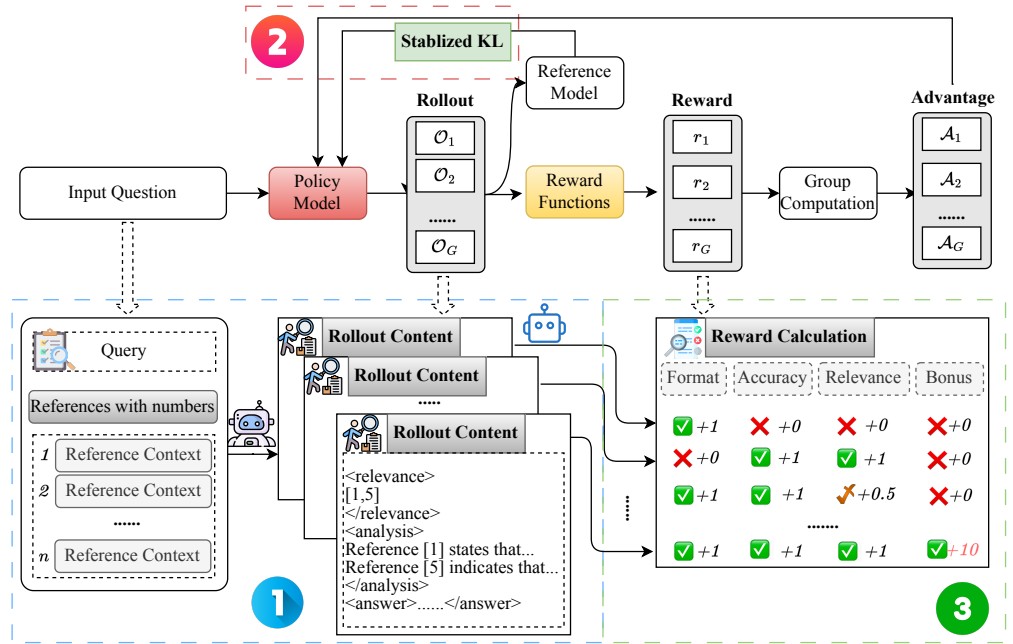

Figure 1: Overview of the **A**daptive-**R**ewarded **E**vidence **N**avigation **A**gent (ARENA) framework. The system is composed of three key components: ① **Structured Protocol**, where the policy model generates a multi-part response including selected evidence, reasoning analysis, and a final answer; ② **KL Stabilization**, we modify the KL divergence estimator to improve stability; ③ **Adaptive Reward Calculation**, where the model outputs are evaluated on four axes—format, accuracy, relevance, and bonus—to provide interpretable and fine-grained training signals.

## 3 ARENA: ADAPTIVE-REWARDED EVIDENCE NAVIGATION AGENT

We introduce **A**daptive-**R**ewarded **E**vidence **N**avigation **A**gent (**ARENA**), a novel reinforcement learning framework for the training of RAG generators. ARENA enhances the reasoning capabilities of LLMs and improves decision transparency, enabling more interpretable and reliable deployment in knowledge-intensive domains. As shown in Figure 1, ARENA consists of three key components: (1) **Structured protocol**, where the model produces evidence-grounded multi-part responses; (2) **KL stabilization**, we stabilized KL regularization for policy refinement; and (3) **Adaptive reward calculation**, which provides interpretable feedback across multiple dimensions. We will introduce these modules in detail in the following subsections.

### 3.1 FRAMEWORK FORMALIZATION

Given a natural language question $q$ and a set of $k$ retrieved reference passages $\mathcal{C} = \{c_1, c_2, ..., c_k\}$, the ARENA generator produces a structured response with three clearly defined components:

- A set of selected reference indices $\mathcal{I} \subseteq \{1, 2, ..., k\}$ indicating which passages are used.
- A reasoning trace $\mathcal{Z}$ that synthesizes the selected passages to generate the final answer.
- A concise final answer $\mathcal{O}$ derived from the reasoning trace.

This defines the generator as a mapping:

$$(q, \mathcal{C}) \mapsto (\mathcal{I}, \mathcal{Z}, \mathcal{O})$$

To support training and evaluation, we construct a structured prompt format as shown in Table 1 (Figure 1 Module 1). Notably, we introduce an explicit `<relevance>` part to enforce interpretable evidence selection, which is critical for transparent multi-hop reasoning. This part requires the model to generate only the reference indices it relies on, making the reasoning process externally verifiable.

Table 1: ARENA prompt template. The `<relevance>` part is the key to encourage explicit and auditable evidence selection. The placeholders {question} and {references} are filled in at runtime.

---

```
A conversation between User and Assistant.  The user asks a question
and gives some references.  The assistant should answer the question
based on the references.

User's input will always contain:
<question>[ the question to answer ]</question>
<references>[ references starting with numbers ]</references>

Assistant's response must contain EXACTLY three sections:
<relevance> [list ONLY reference numbers that provide useful information in square brackets, e.g.
[1,5]]</relevance>
<analysis>[ combine information from relevant references to build the answer. Explicitly mention
which references support each claim ]</analysis>
<answer>[ answer with ONLY a short phrase or single word. no explanations ]</answer>
```

**User:**
```
<question>{question}</question>
<references>{references}</references>
```

---

Each generation component serves a specific role/function: the `<relevance>` part ensures explicit evidence grounding, the `<analysis>` part compels the LLM to articulate its reasoning based on the selected references, and the `<answer>` part provides a concise and definitive result. Together, this format enforces traceable generation and supports external auditing, aligning with ARENA's goals of enhancing reasoning quality and interpretability.

### 3.2 REINFORCEMENT LEARNING WITH STABILITY ENHANCEMENTS

In complex reasoning tasks, supervised fine-tuning (SFT) often falls short due to the lack of high-quality, annotated reasoning traces (Chu et al., 2025). Recently, reinforcement learning (RL) has emerged as a powerful alternative to enhance performance through self-improvement. Motivated by this, ARENA employs reinforcement learning to activate and refine the reasoning capabilities of instruction-tuned models to enhance the RAG generation ablities of them.

BRIEF REVIEW OF GROUP RELATIVE POLICY OPTIMIZATION (GRPO)

GRPO (Shao et al., 2024) is an advanced policy optimization algorithm that improves over Proximal Policy Optimization (PPO) (Schulman et al., 2017) by eliminating the need for a critic model. For each training instance $q$, the policy $\pi_\theta$ generates $G$ outputs $\{o_i\}_{i=1}^G$. Each output is scored using the task-specific reward function (detailed in Section 3.3), and a group-wise normalized advantage is computed as: $\hat{A}_{i,t} = \frac{r_i - \text{mean}(\{r_1, r_2, \cdots, r_G\})}{\text{std}(\{r_1, r_2, \cdots, r_G\})}, \quad r_i = \sum_{j=1}^N w_j \cdot R_j(o_i \mid q),$ .

let $\ell_{i,t}(\theta) = \min\left[\frac{\pi_\theta(o_{i,t}|q,o_{i,<t})}{\pi_{\theta_{old}}(o_{i,t}|q,o_{i,<t})}\hat{A}_{i,t}, \text{clip}\left(\frac{\pi_\theta(o_{i,t}|q,o_{i,<t})}{\pi_{\theta_{old}}(o_{i,t}|q,o_{i,<t})}, 1-\epsilon, 1+\epsilon\right)\hat{A}_{i,t}\right]$,

The final GRPO objective is:

$$\mathcal{J}_{GRPO}(\theta) = \mathbb{E}_{q,\{o_i\}\sim\pi_{\theta_{old}}}\left[\frac{1}{G}\sum_{i=1}^G \frac{1}{|o_i|}\sum_{t=1}^{|o_i|}\left(\ell_{i,t}(\theta) - \beta\,\mathbb{D}_{KL}(\pi_\theta \,\|\, \pi_{ref})\right)\right].$$

KL DIVERGENCE STABILIZATION

In standard GRPO, the KL divergence is instantiated via the $k_3$ **estimator** (Schulman, 2020). Let

$$r(q, o_i) = \frac{\pi_{\text{ref}}(o_{i,<t} \mid q)}{\pi_\theta(o_{i,<t} \mid q)}.$$

The $k_3$ estimate of the KL is

$$L_{k_3}(\theta) = \mathbb{E}_{q,o_i}[\,r(q, o_i) - \log r(q, o_i) - 1\,].$$

In this work we instead adopt the $k_2$ **estimator**:

$$L_{k_2}(\theta) \;=\; \frac{1}{2}\,\mathbb{E}_{q,o_i}\Big[\big(\log r(q,o_i)\big)^2\Big].$$

Compared with $k_3$, the $k_2$ estimator preserves non-negativity and, crucially, offers *lower variance* and *gradient equivalence in expectation* to the KL gradient; moreover, it provides *symmetric* penalties around $r=1$ and exhibits improved stability when $\pi_\theta$ deviates from $\pi_{\text{ref}}$. For these reasons, we use $k_2$ in our GRPO training. **Detailed theoretical analysis and proof can be found in Appendix B**.

### 3.3 Adaptive Reward Design

While recent math-oriented studies demonstrated that simple **format** and **accuracy** rewards suffice to activate reasoning capabilities in GRPO, we find that such task-agnostic metrics fall short in capturing the nuanced objectives of RAG generation. To this end, we propose a set of task-specific, interpretable reward functions that provide fine-grained supervision aligned with reasoning quality and evidence traceability, namely:

**Format Reward** $R_{\text{format}}(o_i \mid q)$**.**  Outputs that match the expected structural format—consisting of `<relevance>`, `<analysis>`, and `<answer>` parts in order—receive a reward of 1; others 0.

**Accuracy Reward** $R_{\text{accuracy}}(o_i \mid q)$**.**  To evaluate the correctness of the final answer, we apply normalized Exact Match. The output from the `<answer>` field is lowercased, stripped of punctuation and articles, and compared to the gold answer. If the normalized strings match exactly, the model receives a reward of 1; otherwise 0. Exact Match is more effective than F1/LLM judgment here.

**Relevance Reward** $R_{\text{relevance}}(o_i \mid q)$**.**  This reward measures whether the model correctly identifies the supporting evidence. The predicted reference indices from the `<relevance>` section are compared with ground truth. A full match yields 1 point, partial overlap yields 0.5, and no overlap yields 0. This encourages models to explicitly ground their reasoning in relevant sources.

**Bonus Reward** $R_{\text{bonus}}(o_i \mid q)$**.**  To promote holistic reasoning behavior, we add a high-value bonus reward when the model simultaneously satisfies all three criteria above. If the format, accuracy, and relevance rewards are all 1, an additional bonus of 10 is assigned; otherwise, the bonus is 0. This reinforces complete, well-aligned outputs.

The final reward used to train the policy model is computed as the sum of the four components and no reweighting strategy/training is adopted:

$$r_i = R_{\text{format}}(o_i \mid q) + R_{\text{accuracy}}(o_i \mid q) + R_{\text{relevance}}(o_i \mid q) + R_{\text{bonus}}(o_i \mid q),$$

where all weights are set to 1 by default. This combined signal balances structural, semantic, and evidential quality in a unified reward framework.

## 4 Experimental Setups

### 4.1 Datasets

We train and evaluate our method on three widely-used multi-hop QA benchmarks: **HotpotQA** (Yang et al., 2018), **2WikiMultiHopQA**(2Wiki) (Ho et al., 2020), and **MuSiQue** (Trivedi et al., 2022). Following prior work (Song et al., 2025a; Wang et al., 2025a; Jin et al., 2025), we randomly sample a total of 25,000 examples from the original training sets for training, 500 examples from each dataset's test set for evaluation. Detailed statistics and preprocessing steps are provided in Appendix C.

### 4.2 Evaluation Metrics

We assess model performance using three standard multi-hop QA metrics (Yang et al., 2018; Gao et al., 2023): normalized **Exact Match** (EM), **F1 score** (F1), and **LLM-as-a-Judge** (LJ) (Zheng et al., 2023). EM and F1 measure surface-level string overlap with the gold answer, while LJ uses

Table 2: Main experimental results across three datasets and three evaluation metrics. Models are grouped by training methodology. **Bold** indicates the best score; underline indicates the second-best. Metrics: EM = Exact Match(%), F1 = F1 score(%), LJ = LLM-as-a-Judge(%).

| Model | HotpotQA | | | 2Wiki | | | MuSiQue | | |
|---|---|---|---|---|---|---|---|---|---|
| | EM | F1 | LJ | EM | F1 | LJ | EM | F1 | LJ |
| **Prompt-based models** | | | | | | | | | |
| Llama3.1-8B-Instruct | 52.8 | 67.6 | 73.8 | 39.8 | 47.8 | 48.4 | 24.8 | 37.0 | 32.4 |
| Qwen2.5-7B-Instruct | 48.4 | 62.8 | 66.0 | 33.4 | 42.4 | 41.2 | 25.2 | 35.4 | 30.6 |
| DeepSeek-R1-Distill-Qwen-7B | 33.2 | 48.7 | 71.2 | 29.0 | 40.7 | 65.8 | 11.6 | 18.4 | 27.8 |
| Qwen3-8B | 58.2 | 71.9 | 76.8 | 65.2 | 72.7 | 78.4 | 33.6 | 39.7 | 39.4 |
| SuRe-GPT-4o | 49.0 | 69.5 | 74.2 | 49.0 | 60.7 | 63.2 | 19.4 | 30.7 | 32.4 |
| SuRe-Qwen-7B | 50.4 | 64.8 | 69.6 | 36.4 | 44.8 | 44.0 | 14.6 | 23.3 | 22.6 |
| self-ask-GPT-4o | 40.8 | 57.7 | 66.2 | 43.0 | 56.5 | 61.4 | 16.2 | 30.5 | 34.6 |
| **SFT-based models** | | | | | | | | | |
| Qwen-7B-SFT-direct | 49.8 | 62.8 | 66.2 | 54.0 | 60.9 | 62.0 | 16.8 | 25.4 | 21.4 |
| Qwen-7B-SFT-reasoning | 40.0 | 53.2 | 59.9 | 53.6 | 61.5 | 62.2 | 11.2 | 17.2 | 17.4 |
| SimpleDeepSearcher-7B | 50.0 | 63.3 | 68.6 | 63.2 | 70.7 | 74.8 | 25.8 | 34.7 | 34.4 |
| **RL-based models** | | | | | | | | | |
| Naive-GRPO | 53.4 | 66.2 | 73.2 | 62.0 | 68.4 | 72.0 | 33.2 | 43.5 | 42.0 |
| R1-Searcher-7B | 59.0 | 73.1 | 79.1 | 63.2 | 70.4 | 73.0 | 25.0 | 37.0 | 32.7 |
| Search-R1-7B | 53.2 | 68.5 | 76.6 | 51.8 | 61.7 | 66.0 | 30.6 | 40.8 | 43.2 |
| ZeroSearch-7B | 51.8 | 64.6 | 70.6 | 43.6 | 51.7 | 53.2 | 27.0 | 36.4 | 34.2 |
| ReSearch-7B | 57.8 | 72.9 | 77.8 | 58.6 | 65.8 | 68.4 | 30.0 | 39.8 | 40.7 |
| **Ours (ARENA)** | | | | | | | | | |
| **ARENA-Llama3.1-8B** | 55.2 | 70.3 | 78.0 | 62.2 | 71.0 | 73.5 | 35.8 | 44.5 | 46.6 |
| **ARENA-Qwen2.5-7B** | **62.8** | **76.2** | **81.2** | **66.0** | **75.2** | **78.6** | **40.0** | **52.0** | **50.8** |

GPT-4o (OpenAI, 2024) (version 2024-11-20) to provide a semantic judgment of answer correctness. The implementation details of these three metrics are provided in Appendix D. In addition to these, we introduce two auxiliary evaluation metrics: **Format score** and **Relevance score**, which assess instruction adherence and citation accuracy. These scores follow the same evaluation criteria as used in the reward computation during training (see Section 3.3).

## 4.3 BASELINES

We compare our method against a comprehensive set of baselines across three categories: **(1) Prompt-based models:** we include Qwen2.5-7B-Instruct (Yang et al., 2024), Llama3.1-8B-Instruct (Grattafiori et al., 2024), DeepSeek-R1-Distill-Qwen-7B (Guo et al., 2025), and Qwen3-8B (Yang et al., 2025), which directly answer questions with references[2]. We also consider SuRe (Kim et al., 2024), which enhances performance via summarized retrieval, and self-ask (Press et al., 2023), which decomposes complex questions into sub-questions and answers them iteratively. **(2) SFT-based models:** we implement two variants: SFT-Direct, which directly generates the final answer, and SFT-Reasoning, which produces structured thoughts and answers. Implementation details are provided in Appendix E. We also include SimpleDeepSearcher (Sun et al., 2025b), which trains models using automatically generated reasoning and retrieval trajectories. **(3) RL-based models:** we implement a Naive-GRPO baseline using the original GRPO algorithm (Shao et al., 2024), where models are trained to output structured reasoning with format and accuracy rewards. In addition, we evaluate against four recent RL-based RAG frameworks: R1-Searcher (Song et al., 2025a), Search-R1 (Jin et al., 2025), ZeroSearch (Sun et al., 2025a), and ReSearch (Chen et al., 2025a), which all aim to teach the model to actively retrieve and reason through reinforcement learning. For consistency, we use their released checkpoints and replace their retrieved contexts with those provided by datasets.

---

[2]For reasoning-capable models, the internal thinking process is implicitly conducted during inference.

Table 3: Evaluation of **Format** and **Relevance** scores (%) for ARENA-7B and Qwen2.5-7B-Instruct on three datasets. ARENA-7B shows consistent and substantial improvements across all datasets.

| Model | HotpotQA | | 2Wiki | | MuSiQue | |
|---|---|---|---|---|---|---|
| | Format | Relevance | Format | Relevance | Format | Relevance |
| Qwen2.5-7B-Instruct | 91.6 | 68.3 | 97.8 | 67.3 | 80.2 | 48.4 |
| ARENA-7B | **100.0** | **91.3** | **100.0** | **90.8** | **100.0** | **59.5** |

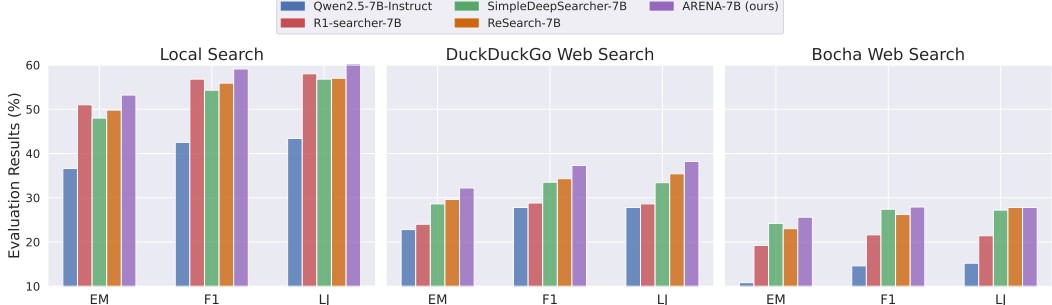

Figure 2: Evaluation results on 2Wiki under real-world local search and web search scenarios. ARENA-7B consistently achieves the best performance across all metrics and search modalities, demonstrating its superior generalization and reasoning capabilities in multi-hop question answering.

## 4.4 IMPLEMENTATION DETAILS

We conduct **ARENA** on `Qwen2.5-7B-Instruct` and `Llama3.1-8B-Instruct`. Our framework is implemented based on the `Open-R1` (Face, 2025). During inference, we concatenate all retrieved paragraphs provided by the dataset in their original order. All models are trained using 8 NVIDIA A100-80G GPUs, with 7 allocated for policy optimization and 1 dedicated to rollout inference via a vLLM (Kwon et al., 2023) engine. More training details are introduced in Appendix F.

## 5 EXPERIMENTAL RESULTS

### 5.1 MAIN RESULTS

As shown in Table 2, ARENA-Qwen2.5-7B (ARENA-7B) achieves the best performance across all datasets and metrics, consistently outperforming all baselines of similar scale. ARENA-Llama3.1-8B also outperforms the original Llama3.1-8B-Instruct significantly, validating the effectiveness of our training method across distinct backbones.

**Transparency.** One key goal of ARENA is to improve the traceability and interpretability of RAG generator. We quantitatively evaluate the models before and after RL training by Format score and Relevance score, which measure instruction-following behavior and citation accuracy. As shown in Table 3, ARENA-7B significantly outperforms Qwen2.5-7B-Instruct in both metrics across all datasets. These results indicate that our method greatly enhances the transparency of the generated responses, enabling more faithful and controllable reasoning.

**Stability.** We not only provide a theoretical analysis showing the superiority of the $k_2$ estimator over the commonly used $k_3$ estimator, but also validate this claim empirically. As shown in the KL divergence curves during training (see Appendix G), the stabilized variant based on $k_2$ maintains smoother KL values and avoids extreme spikes, confirming its effectiveness in preventing degenerate updates during RL optimization.

### 5.2 GENERALIZATION ABILITY

Table 4: Evaluation of ARENA-7B under different training dataset combinations. "+WM" denotes training on 2**W**iki + **M**uSiQue, "+HM" on **H**otpotQA + **M**uSiQue, and "+HWM" on all three datasets. Results are reported on all three test sets to assess cross-domain generalization.

| Model | HotpotQA | | | 2Wiki | | | MuSiQue | | |
|---|---|---|---|---|---|---|---|---|---|
| | EM | F1 | LJ | EM | F1 | LJ | EM | F1 | LJ |
| ARENA-7B + WM | 56.8 | 71.1 | 76.4 | 64.2 | 73.2 | 73.4 | 39.4 | 50.4 | 48.6 |
| ARENA-7B + HM | 59.8 | 74.1 | 79.6 | 57.8 | 67.1 | 68.0 | **40.4** | 51.9 | 47.8 |
| ARENA-7B + HWM | **62.8** | **76.2** | **81.2** | **66.0** | **75.2** | **77.4** | 40.0 | **52.0** | **50.8** |

We demonstrate that our method generalizes well across unseen tasks, models, and retrieval settings.

**Dataset generalization.** As shown in Table 4, we evaluate ARENA-7B trained on different combinations of datasets to assess zero-shot transfer across domains. In both cases, the model performs strongly on the held-out dataset, indicating that the reasoning capabilities learned by ARENA generalize well across different data distributions.

**Retrieval setting generalization.** While our primary experiments use dataset-provided references, we also evaluate ARENA under realistic retrieval settings. We conduct experiments using local search from Wikipedia (Petroni et al., 2021), and web search via DuckDuckGo (Hands, 2012) and Bocha (Bochaai). Results in Figure 2 show that ARENA consistently outperforms baselines under all retrieval settings, demonstrating its robustness and effectiveness in real-world scenarios.

**Model and task generalization.** Our method also brings performance gains on smaller models (Qwen2.5-3B-Instruct), and shows strong transferability when applied to a summarization task using a similar RL approach. See Appendix H for details.

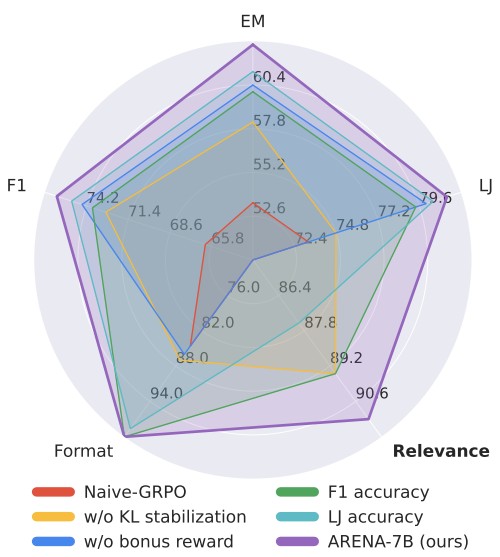

Figure 3: Radar chart comparing six variants across five evaluation metrics on HotPotQA. ARENA shows consistent improvements in output quality and interpretability.

### 5.3 ABLATION STUDY

We conduct an ablation study to evaluate the contribution of each component in our method. As shown in Figure 3, we compare six model variants on HotpotQA across five metrics: EM, F1, LJ, Format score, and Relevance score.

**Naive-GRPO** represents the basic GRPO setup. **w/o KL stabilization** removes our improved estimator for KL divergence. **w/o bonus reward** disables the bonus term in the adaptive reward design (see Appendix I for further analysis). **F1 accuracy** and **LJ accuracy** use F1 score and LLM-as-a-Judge score to compute the accuracy reward.

Results show that **ARENA-7B** consistently outperforms all other variants across all evaluation dimensions, demonstrating its advantage in both effectiveness (EM, F1, LJ) and transparency (Format, Relevance). These results confirm the value of each component in our design.

### 5.4 FURTHER ANALYSES

**ARENA vs. Advanced Commercially-developed Models.** Our experiments indicate that 7B-scale models exhibit a significant gap in RAG generation capability compared to advanced closed-source models. However, after reinforcement learning, our ARENA-7B achieves performance that is comparable to these commercially-developed models across multiple metrics (See Appendix J).

---

🚪 **Input:**

**Q:** What party was the secretary of commerce and housing under the 42 Governor of Kansas?

**Reference 1~Reference 5.** David "Dave" Lopez......**Omit 392 words**

**Reference 6.** Joan Finney (February 12, 1925 – July 28, 2001), served as the 42nd Governor of Kansas from 1991 to 1995.

**Reference 7.** William Thornton is an American politician from......**Omit 48 words**

**Reference 8.** Robert G. "Bob" Knight (born July 31, 1941) was the Republican mayor of Wichita, Kansas for seven terms. He first ran for office, Wichita City Council, in 1979. He also served under Democratic Governor Joan Finney as Kansas Secretary of Commerce and Housing. He is generally known as "Bob Knight."

**Reference 9.** Natalie Shirley (born 1957) is an American lawyer......**Omit 88 words**

**Reference 10.** The Kansas Department of Commerce is a......**Omit 55 words**

---

📋❓ **Naive GRPO**

<think>The question asks about the party of the secretary of commerce and housing under the 42nd Governor of Kansas\nThe reference information states that Joan Finney was the 42nd Governor of Kansas from 1991 to 1995\nThe references do not provide specific information about the party of the secretary of commerce and housing under Joan Finney</think>
<answer>Democratic</answer> ❌

---

🔍 **ARENA (Ours)**

<relevance>[6,8]</relevance>
<analysis>Governor Joan Finney, the 42nd Governor of Kansas, served from 1991 to 1995 [6]. Under her administration, Robert G. "Bob" Knight served as the Kansas Secretary of Commerce and Housing [8]. Therefore, the party affiliation of the secretary of commerce and housing under the 42nd Governor of Kansas can be inferred from Knight's party affiliation. Reference [8] also confirms that Knight was appointed by the Democratic Governor Joan Finney, implying that Knight was a member of the Republican party.</analysis>
<answer>Republican</answer> ✓

Figure 4: A Case Study of Naive GRPO and ARENA on the HotpotQA. ARENA correctly extracts relevant paragraphs, provides a structured explanation, and delivers the right answer.

**Accuracy Across Reasoning Hops.** We further investigate how reasoning complexity affects performance by grouping questions based on their required number of hops. Our analysis shows that questions requiring fewer hops are consistently easier to answer and retrieve relevant evidence. See Appendix K for detailed results and discussion.

**Case Study.** Figure 4 illustrates a representative case while Naive GRPO produces a valid-looking but wrong answer, ARENA identifies the correct references and generates a transparent reasoning.

## 6 CONCLUSION

We propose **ARENA**, an **A**daptive-**R**ewarded **E**vidence **N**avigation **A**gent that enhances the generator component of RAG systems via reinforcement learning with task-specific rewards. ARENA enables LLMs to explicitly link selected references, reasoning traces, and final answers. We further introduce a quantitative evaluation of citation relevance, making decision paths both auditable and measurable. We also study KL estimators for RAG training, develop stabilization strategies, and provide supporting empirical evidence of improved training stability and output quality. Extensive experiments on three multi-hop QA benchmarks show 10–30% accuracy gains over open-source models of similar scale, together with strong generalization across datasets and tasks. By combining transparent, evaluable reasoning with robust optimization, ARENA advances both interpretability and effectiveness for real-world RAG applications.

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

## A  USE OF LARGE LANGUAGE MODELS

We use Large Language Models to aid or polish writing.

## B  THEORETICAL ANALYSIS OF KL ESTIMATORS

Building on derivations from prior work Schulman (2020); Liu et al. (2025), we aim to show that the $k_2$ estimator yields more stable training than the $k_3$ estimator.

**Notation.** Let $q \sim \mathcal{D}$ denote inputs, $o \sim \pi_\theta(\cdot \mid q)$ denote samples from the current policy, and $\pi_{\mathrm{ref}}$ be a fixed reference policy. Define

$$r(q, o_i) \;=\; \frac{\pi_{\mathrm{ref}}(o_{i,<t} \mid q)}{\pi_\theta(o_{i,<t} \mid q)} \,.$$

**KL divergence and its gradient.**

$$\mathbb{D}_{\mathrm{KL}}\big(\pi_\theta(\cdot \mid q) \,\|\, \pi_{\mathrm{ref}}(\cdot \mid q)\big) \;=\; \mathbb{E}_{o \sim \pi_\theta}\left[\log \frac{\pi_\theta(o_{i,<t} \mid q)}{\pi_{\mathrm{ref}}(o_{i,<t} \mid q)}\right] ,$$

$$-\nabla_\theta \mathbb{D}_{\mathrm{KL}} \;=\; -\mathbb{E}_{o \sim \pi_\theta}\left[\log \frac{\pi_\theta(o_{i,<t} \mid q)}{\pi_{\mathrm{ref}}(o_{i,<t} \mid q)} \, \nabla_\theta \log \pi_\theta(o_{i,<t} \mid q)\right]$$

$$\;=\; -\mathbb{E}[-\log r(q, o_i) \, \nabla_\theta \log \pi_\theta(o_{i,<t} \mid q)] \,.$$

$k_2$ **estimator.**

$$L_{k_2}(\theta) \;=\; \frac{1}{2}\,\mathbb{E}_{q,o}\Big[\big(\log r(q,o_i)\big)^2\Big]\,,$$

$$\nabla_\theta L_{k_2} \;=\; \mathbb{E}_{q,o}[\log r(q,o_i)\,\nabla_\theta \log r(q,o_i)]\,,$$

$$\nabla_\theta \log r(q,o_i) \;=\; \nabla_\theta(\log \pi_{\mathrm{ref}}(o_{i,<t}\mid q) - \log \pi_\theta(o_{i,<t}\mid q)) \;=\; -\nabla_\theta \log \pi_\theta(o_{i,<t}\mid q)\,,$$

$$-\nabla_\theta L_{k_2} \;=\; \mathbb{E}_{q,o}[\log r(q,o_i)\,\nabla_\theta \log \pi_\theta(o_{i,<t}\mid q)]\,.$$

$k_3$ **estimator.**

$$L_{k_3}(\theta) \;=\; \mathbb{E}_{q,o}[r(q,o_i) - \log r(q,o_i) - 1]\,,$$

$$\nabla_\theta L_{k_3} \;=\; \mathbb{E}_{q,o}[\nabla_\theta r(q,o_i) - \nabla_\theta \log r(q,o_i)]\,,$$

$$\nabla_\theta r(q,o_i) \;=\; r(q,o_i)\,\nabla_\theta \log r(q,o_i) \;=\; -r(q,o_i)\,\nabla_\theta \log \pi_\theta(o_{i,<t}\mid q)\,,$$

$$-\nabla_\theta L_{k_3} \;=\; \mathbb{E}_{q,o}\big[(r(q,o_i) - 1)\,\nabla_\theta \log \pi_\theta(o_{i,<t}\mid q)\big]\,.$$

**Comparison.**
(i) Gradient equivalence: $-\nabla_\theta L_{k_2} = \mathbb{E}[-\log r\,\nabla_\theta \log \pi_\theta]$ matches $-\nabla_\theta \mathbb{D}_{\mathrm{KL}}$ in expectation, whereas $-\nabla_\theta L_{k_3} = \mathbb{E}[(r-1)\,\nabla_\theta \log \pi_\theta] \neq -\nabla_\theta \mathbb{D}_{\mathrm{KL}}$.
(ii) Bias under large deviation: when $\pi_\theta$ and $\pi_{\mathrm{ref}}$ diverge in later training (e.g., $r \gg 1$ or $r \ll 1$), the $(r-1)$ weighting in $k_3$ induces a large bias relative to the KL gradient, yielding mis-scaled updates.
(iii) Symmetry of penalties: the $k_2$ loss $\frac{1}{2}(\log r)^2$ penalizes positive and negative deviations symmetrically around $r=1$, aligning with the goal of RL regularization for LLMs; by contrast, $k_3$ is asymmetric and can over- or under-penalize depending on deviation direction.

## C  DATASETS, PREPROCESSING, AND EVALUATION SETUP

We follow the preprocessing pipeline from PIKE-RAG (Wang et al., 2025a) to format the train and dev splits for all datasets. Each example includes:

- A natural language question.

- A list of 10-20 retrieved paragraphs from Wikipedia.

- A gold-standard answer (often as a list of acceptable synonyms).

- A set of `supporting_facts`, which mark the sentences essential for answering the question.

To construct the `<references>` field, we concatenate all retrieved paragraphs in their original order with paragraph-level IDs (1 to $n$). To supervise relevance prediction and reasoning, we use the `supporting_facts` annotations to locate which paragraphs contain the necessary sentences. These are mapped to their paragraph indices to produce a ground-truth list of relevant reference IDs.

Following prior work (Song et al., 2025a; Wang et al., 2025a; Jin et al., 2025), we randomly sample 10,000 training examples from HotpotQA, 10,000 from 2WikiMultiHopQA, and 5,000 from MuSiQue, totaling 25,000 training instances. For evaluation, we sample 500 development examples from each dataset. Dataset statistics are summarized in Table 5.

**Answer Format.**  The answer labels are often represented as a list of acceptable values (e.g., synonyms, numbers, yes/no), and models are considered correct if their normalized output matches any item in the list. This setting is used consistently across all automatic and LLM-based evaluations.

## D  DETAILS OF EVALUATION METRICS

We use three evaluation metrics in our experiments: Exact Match (EM), F1 Score (F1), and LLM-as-a-Judge (LJ). Below we describe the implementation details of each.

Table 5: Detailed dataset statistics. Hop counts are derived from `supporting_facts` fields.

| Dataset | Split | Data Size | # Paragraphs | 1-hop | 2-hop | 3-hop | 4-hop | Avg. Hops |
|---------|-------|-----------|--------------|-------|-------|-------|-------|-----------|
| HotpotQA | Train | 10,000 | 10 | 3 | 9977 | 20 | 0 | 2.00 |
| HotpotQA | Test | 500 | 10 | 0 | 500 | 0 | 0 | 2.00 |
| 2Wiki | Train | 10,000 | 10 | 7 | 7839 | 3 | 2151 | 2.39 |
| 2Wiki | Test | 500 | 10 | 0 | 406 | 0 | 94 | 2.19 |
| MuSiQue | Train | 5,000 | 20 | 0 | 3595 | 1107 | 298 | 2.85 |
| MuSiQue | Test | 500 | 20 | 0 | 251 | 153 | 96 | 2.73 |
| **All Train** | – | 25,000 | – | 10 | 21,411 | 1130 | 2449 | 2.30 |

## D.1 EXACT MATCH AND NORMALIZATION

The Exact Match (EM) metric determines whether the model's answer exactly matches any of the reference answers after normalization. The normalization process includes:

- Lowercasing the text.
- Removing punctuation and articles (*a*, *an*, *the*).
- Replacing underscores with spaces.
- Collapsing multiple spaces into a single space.

This ensures robustness to minor formatting or surface-level differences. A score of 1.0 is returned for an exact match; otherwise, 0.0.

## D.2 F1 SCORE CALCULATION

The F1 score is computed at the token level based on normalized answers. For each reference answer, we calculate the precision and recall between the predicted and ground truth tokens, and derive the F1 score as their harmonic mean. The maximum F1 score across all ground truths is taken as the final score for a given prediction.

## D.3 LLM-AS-A-JUDGE EVALUATION

To evaluate semantic correctness that may not be captured by surface-level metrics, we adopt a GPT-based LLM-as-a-Judge (LJ) method. For each question, the model's generated response and the reference answers are embedded into a prompt and passed to GPT-4o (version 2024-11-20). The model is instructed to answer `"yes"` if the response is semantically consistent with any of the correct answers (including paraphrases or synonyms), and `"no"` otherwise.

The prompt template is shown in Table 6:

Table 6: Prompt template used in LLM-as-a-Judge evaluation. The placeholders {correct_answer} and {model_response} are replaced at runtime.

```
*************Consider a knowledge Q&A RAG task to test the capability
of a testing model, the correct answer list is:*************
{correct_answer}

*************Here is the model's response:*************
{model_response}

*************Please check if the model's answer is correct.  As long
as the model's answer hits any item (or synonym) in the correct
answer list, it can be considered correct.  You only need to answer
"yes" or "no".*************
```

When calling the GPT-4o API, we set the temperature to `0.0` to ensure deterministic judgment behavior.

## E    DETAILS OF SUPERVISED FINE-TUNING BASELINES

To highlight the benefits of reinforcement learning, we design two supervised fine-tuning (SFT) baselines:

**SFT-direct.**    This variant trains the model to generate the final answer directly. During training, we concatenate the input question with all retrieved paragraphs (in their original order) and instruct the model to output a short answer without any intermediate reasoning or evidence selection. This setup resembles conventional instruction-tuning on flat context.

**SFT-reasoning.**    This variant follows a two-stage pipeline to teach the model structured reasoning. Specifically, for each training instance, we collect:

- Question $q$

- Retrieved reference set $\mathcal{C}$

- Ground-truth relevant reference IDs $\mathcal{I}$

- Ground-truth final answer $\mathcal{O}$

These components are fed into the backbone model using a reasoning prompt shown in Table 7, prompting it to generate a coherent reasoning trace. The output reasoning trace is then integrated into the structured format described in Section 3.1:

$$(q, \mathcal{C}) \rightarrow (\mathcal{I}, \mathcal{Z}, \mathcal{O})$$

This structured format is then used for supervised fine-tuning with full output supervision.

Table 7: Prompt format for collecting structured reasoning traces in SFT-reasoning. The model is instructed to derive the answer using only specified relevant references.

```
Generate a thinking process showing how to derive the given answer
using ONLY the specified relevance IDs from provided references.
Question: {question}
References: {references}
Relevance IDs: {relevance_ids}
Answer: {answer}
```

## F    DETAILS OF TRAINING CONFIGURATION

We train all models using the `Open-R1` reinforcement learning framework with DeepSpeed ZeRO Stage 2 optimization and bfloat16 mixed-precision. The total batch size is 256, learning rate is set to 3e-6. We generate 7 rollout samples per input for reward estimation. We set temperature = 0.9 during rollout, KL coefficient $\beta = 0.04$, number of iterations per batch $\mu = 1$, and clipping parameter $\epsilon = 0.2$. Our experiments were conducted using 8×A100-80G GPUs for approximately 20 hours.

## G    EMPIRICAL ANALYSIS OF KL ESTIMATORS

To examine the effect of our stabilized KL divergence formulation (see Section 3.2), we compare its training dynamics with the standard KL estimator. As shown in Figure 5, the stabilized variant exhibits smooth and bounded updates throughout training, while the original estimator suffers from catastrophic spikes. This confirms that our modified KL term not only improves theoretical stability but also provides better empirical convergence in practice.

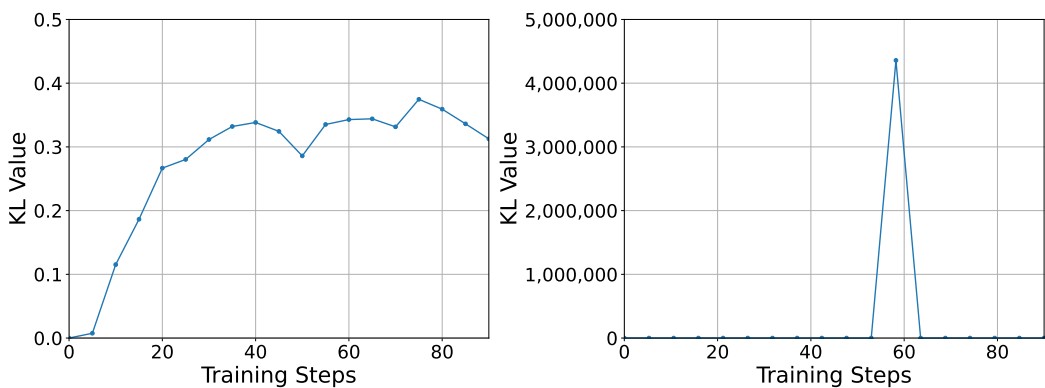

Figure 5: Training KL divergence curves for stabilized vs. unstabilized variants. ARENA's stabilized KL (left) leads to smoother convergence, while the standard estimator (right) suffers from extreme spikes.

# H  CROSS-TASK AND CROSS-SCALE GENERALIZATION

## H.1  SMALL-SCALE MODEL EXPERIMENTS

To explore the feasibility of our method under more resource-constrained settings, we conducted experiments using **Qwen2.5-3B-Instruct**, a smaller backbone model. The evaluation results on the **HotpotQA** dataset are shown in Table 8.

Table 8: Performance of ARENA on the 3B-scale model.

| Model | EM | F1 | Format | Relevance |
|---|---|---|---|---|
| Qwen2.5-3B-Instruct | 37.2 | 50.0 | 90.2 | 47.5 |
| ARENA-3B (ours) | **52.8** | **67.6** | **100.0** | **81.2** |

ARENA-3B demonstrates substantial improvements in both EM and F1 over the untrained version, while also enhancing format adherence and citation accuracy.

## H.2  TASK TRANSFER TO SUMMARIZATION

To further test the generality of our approach, we conducted preliminary experiments applying ARENA to a summarization task. Following our discussion, we hypothesized that **task-specific, structured reward functions** can generalize beyond QA.

We collected several summarization datasets from the Super-NaturalInstructions benchmark (Wang et al., 2022), including topic-focused news summarization (Narayan et al., 2018). Using GPT-4o (version 2024-11-20), we annotated relevant source sentences for each reference summary. This enabled us to construct structured triples for RL training: `<relevance> <analysis> <summary>`, analogous to our QA setting.

We trained models on 5,000 examples and evaluated on 500 held-out samples. Results are presented in Table 9.

As shown above, **ARENA-summary-3B achieves the best performance across all ROUGE metrics**, confirming that our reward-guided RL framework effectively extends to summarization. Notably, even ARENA-3B trained on QA improves over the base model, indicating promising cross-task transferability.

Table 9: Summarization performance using structured RL with customized rewards.

| Model | R-1 | R-2 | R-L | R-avg | Relevance |
|---|---|---|---|---|---|
| Qwen2.5-3B-Instruct | 22.6 | 4.3 | 15.3 | 14.1 | – |
| ARENA-3B (RAG) | 23.7 | 4.4 | 16.3 | 14.8 | 46.6 |
| SFT-3B | 25.5 | 5.8 | 17.5 | 16.3 | – |
| Naive-GRPO-3B | 28.9 | 7.2 | 20.8 | 19.0 | – |
| ARENA-summary-3B (ours) | **29.7** | **7.6** | **21.4** | **19.6** | **53.0** |

# I  ABLATION ON THE BONUS REWARD

To assess the contribution of the bonus reward, we conduct a focused ablation where we retain the **format**, **accuracy**, and **relevance** reward components, but remove the **bonus reward** that is normally applied when all three criteria are simultaneously satisfied.

While this ablated model (**ARENA-7B-nobonus**) continues to improve on accuracy metrics (EM and F1), we observe a clear degradation in structural metrics over time. As shown in Table 10, both format and relevance scores decline substantially after step 75, and the final model fails to consistently generate well-structured and attributed outputs. These results demonstrate the crucial role of the **bonus reward** in maintaining output consistency throughout training.

Table 10: Training trajectory of ARENA without the bonus reward (evaluated on HotpotQA).

| Variant | EM | F1 | Format | Relevance |
|---|---|---|---|---|
| ARENA-7B-nobonus_checkpoint-25 | 58.4 | 73.2 | **99.4** | **86.2** |
| ARENA-7B-nobonus_checkpoint-50 | 58.8 | 73.2 | 95.6 | 81.9 |
| ARENA-7B-nobonus_checkpoint-75 | 58.2 | 73.2 | 91.0 | 29.5 |
| ARENA-7B-nobonus (step 95) | **60.4** | **74.5** | 86.0 | 13.6 |

# J  COMPARING ARENA-7B WITH ADVANCED CLOSED-SOURCE REASONING MODELS

To better understand the performance gap between open-source 7B models and leading closed-source resoning models, we conduct pilot experiments across several LLMs under two inference settings: *direct*, where the model answers based solely on the question, and *direct RAG*, where the model receives both the question and retrieved context passages as input.

As shown in Table 11, models such as GPT-4o, OpenAI-o1, and DeepSeek-R1 significantly outperform open 7B models. These results reveal a substantial gap in RAG generation quality and highlight the need to improve the answer generation capabilities of smaller open-source models.

According to Table 12, our ARENA-7B achieves performance comparable to GPT-4o and DeepSeek-R1 across all evaluation metrics, and substantially narrows the gap with OpenAI-o1. This demonstrates the effectiveness of our structured RL training in improving output quality, making smaller open-source models more competitive for multi-hop reasoning.

# K  HOP-LEVEL ACCURACY ANALYSIS

We analyzed model performance across different reasoning hop levels. The results are summarized in Table 13. As expected, questions with fewer reasoning hops generally yield higher accuracy and relevance, while those with more hops—particularly in MuSiQue—pose significant challenges. These results highlight the difficulty of multi-hop retrieval and reasoning.

Table 11: Performance of vanilla models under two inference strategies: *direct* and *direct RAG*. All models are evaluated using identical retrieved contexts. **Bold** indicates the highest value per column. Metrics: EM = Exact Match, F1 = F1 score, LJ = LLM-as-a-Judge.

| Model | HotpotQA | | | 2Wiki | | | MuSiQue | | |
|---|---|---|---|---|---|---|---|---|---|
| | EM | F1 | LJ | EM | F1 | LJ | EM | F1 | LJ |
| **Direct** | | | | | | | | | |
| Qwen2.5-7B-Instruct | 18.8 | 24.7 | 26.2 | 22.2 | 25.3 | 23.4 | 4.0 | 9.0 | 6.1 |
| Llama3.1-8B-Instruct | 19.4 | 28.3 | 28.4 | 21.0 | 24.6 | 23.2 | 3.8 | 8.0 | 5.4 |
| DeepSeek-R1-Distill-Qwen-7B | 8.6 | 15.0 | 16.2 | 14.4 | 20.6 | 23.4 | 1.2 | 4.3 | 2.6 |
| Qwen3-8B | 24.8 | 34.4 | 36.7 | 22.0 | 26.3 | 26.4 | 3.0 | 9.2 | 6.0 |
| GPT-4o | 37.8 | 50.0 | 50.0 | 34.2 | 38.6 | 36.6 | 10.0 | 19.2 | 12.4 |
| OpenAI-o1 | 51.0 | 66.7 | 72.6 | 53.0 | 63.4 | 68.4 | 29.2 | 40.4 | 39.2 |
| DeepSeek-R1 | 42.2 | 56.4 | 59.0 | 45.4 | 53.6 | 54.6 | 20.2 | 33.0 | 29.4 |
| **Direct RAG** | | | | | | | | | |
| Qwen2.5-7B-Instruct | 48.4 | 62.8 | 66.0 | 33.4 | 42.4 | 41.2 | 25.2 | 35.4 | 30.6 |
| Llama3.1-8B-Instruct | 52.8 | 67.6 | 73.8 | 39.8 | 47.8 | 48.4 | 24.8 | 37.0 | 32.4 |
| DeepSeek-R1-Distill-Qwen-7B | 33.2 | 48.7 | 71.2 | 29.0 | 40.7 | 65.8 | 11.6 | 18.4 | 27.8 |
| Qwen3-8B | 58.2 | 71.9 | 76.8 | 65.2 | 72.7 | 78.4 | 33.6 | 39.7 | 39.4 |
| GPT-4o | 62.8 | 78.8 | 82.0 | 60.6 | 68.7 | 71.7 | 50.5 | 61.7 | 59.3 |
| OpenAI-o1 | **65.6** | **81.6** | **89.0** | 70.2 | 80.5 | 83.7 | **62.2** | **74.4** | **76.2** |
| DeepSeek-R1 | 61.8 | 78.1 | 83.6 | **72.1** | 80.1 | **83.8** | 55.9 | 70.2 | 70.3 |

Table 12: Comparison between ARENA-7B and leading closed-source models on HotpotQA.

| Model | EM | F1 | LJ |
|---|---|---|---|
| GPT-4o | 62.8 | 78.8 | 82.0 |
| OpenAI-o1 | **65.6** | **81.6** | **89.0** |
| DeepSeek-R1 | 61.8 | 78.1 | 83.6 |
| ARENA-7B | 62.8 | 76.2 | 81.2 |

Table 13: Accuracy of ARENA-7B across different reasoning hop levels.

| Dataset | Hops | Total | Relevance | Answer | All Correct |
|---|---|---|---|---|---|
| HotpotQA | 2 | 500 | 416 / 83.2% | 314 / 62.8% | 272 / 54.4% |
| 2Wiki | 2 | 406 | 360 / 88.7% | 249 / 61.3% | 232 / 57.1% |
| 2Wiki | 4 | 94 | 49 / 52.1% | 81 / 86.2% | 47 / 50.0% |
| MuSiQue | 2 | 251 | 125 / 49.8% | 109 / 43.4% | 76 / 30.3% |
| MuSiQue | 3 | 153 | 4 / 2.6% | 55 / 36.0% | 3 / 2.0% |
| MuSiQue | 4 | 96 | 1 / 1.0% | 36 / 37.5% | 0 / 0.0% |

