# OpenReview forum: "Transparent and Robust RAG: Adaptive-Reward Reinforcement Learning for Decision Traceability"
_ICLR.cc/2026/Conference — ICLR 2026 Conference Withdrawn Submission_

### Official Review · Reviewer_5qa9 · 2025-10-20

**Soundness:** 3
**Presentation:** 2
**Contribution:** 1
**Rating:** 2
**Confidence:** 4

**Summary:**

This paper introduces ARENA, a framework aimed at enhancing the generator component of retrieval-augmented generation (RAG). The authors identify two main shortcomings in existing RAG methods: (1) inadequate emphasis on improving the generator’s reasoning ability, and (2) limited transparency in its decision-making process. ARENA incorporates the following key components: (1) structured generation for more interpretable reasoning, (2) adaptive reward functions tailored to multi-hop question answering, and (3) KL-based stabilization techniques to ensure more stable RL. Experiments show that ARENA has strong generalization to unseen datasets and tasks.

**Strengths:**

1. The paper targets an important practical issue: traceability and interpretability in RAG systems.

2. Incorporating explicit evidence citation and structured output could benefit downstream auditing and reasoning interpretability.

**Weaknesses:**

1. **Heuristic and under-analyzed reward design.**
The adaptive reward shaping is highly ad hoc: the use of a fixed +10 bonus arbitrarily distorts the reward distribution. There is no sensitivity or ablation study to justify the weighting choices, making it unclear whether the reported gains are robust or merely tuned artifacts. Also, it seems that the authors combine several different rewards together in RL, which is incremental in nature.


2. **Conceptual ambiguity around “transparency” and “robustness.”**
These terms are used repeatedly but without precise definition or operationalization. It remains unclear whether the improvements stem from genuinely better reasoning and retrieval grounding, or simply from stronger format adherence due to the structured output template.

3. **Limited technical novelty.**
The paper lacks genuine methodological innovation. All major components—structured output generation, adaptive rewards, and KL stabilization—are incremental variations of well-established ideas from GRPO, DeepSeek-R1, or R1-Searcher. Their combination feels more like an engineering integration rather than a principled advance in RAG optimization.

4. **Narrow and unconvincing experimental scope.**
The evaluation is restricted to simple multi-hop QA. It is also better to use some other datasets that requires knowledge and reasoning, such as browsecomp, simpleQA, GPQA etc., for better evaluation.

**Questions:**

See above.

Missing reference: [1] Tang, Yunhao, and Rémi Munos. "On a few pitfalls in KL divergence gradient estimation for RL." arXiv preprint arXiv:2506.09477 (2025).

This paper also examines the use of K2 and K3 estimators for KL regularization in reinforcement learning. If this submission focuses on the general benefits of using K2 as the regularization term, it would strengthen the work to include additional benchmarks, such as MATH or Code reasoning, to demonstrate broader applicability beyond the current evaluation scope.

---

> ### Author Response · Authors · 2025-11-19
>
> We thank the reviewer for agreeing on the importance of traceability and interpretability in RAG systems, as well as the potential value of structured evidence citation. Our goal is to explore **how transparency-oriented and stability-oriented reinforcement learning can enhance the generator component of RAG**, improving its reasoning quality, evidence usage, and auditability. Below, we address each concern in detail.
>
>
> ---
>
> **W1 — On reward design and the alleged heuristic nature of the bonus reward**
>
> Our reward design is **task-driven rather than heuristic**, and each component directly supervises a key aspect of transparent and traceable RAG reasoning:
>
> * **Relevance reward** provides fine-grained feedback on evidence selection quality.
> * **Format reward** enforces structured and auditable reasoning chains.
> * **Accuracy reward** ensures correctness of the final answer.
> * **Bonus reward** resolves reward ties among partially correct trajectories and incentivizes the rare but essential case of being *simultaneously* correct in format, accuracy, and relevance.
>
> As shown in our ablation study (Appendix I), removing the bonus reward leads to **rapid deterioration of format adherence** during training. This is because, with multiple reward components, many “partially correct” trajectories receive identical total rewards, making it difficult for GRPO’s group-based advantage to prioritize fully correct reasoning paths. The bonus reward is therefore **not an ad-hoc addition**, but a task-aligned mechanism ensuring that fully correct reasoning remains learnable.
>
> To further address the concern about the fixed value of the bonus reward, we include a sensitivity analysis that evaluates bonus magnitudes of **5**, **10**, and **15**. Importantly, the choice of **10** is *not* the result of fine-tuning; instead, it reflects a **safe scale separation**—large enough to stand out from individual reward components (1 point each), yet small enough to maintain optimization stability.
>
> Below we report the performance **deltas** of bonus = 5 and bonus = 15 relative to the default **bonus = 10**. Positive values indicate improvement; negative values indicate decrease.
>
> | Bonus Value | HotpotQA ΔEM / ΔF1 | 2Wiki ΔEM / ΔF1 | MuSiQue ΔEM / ΔF1 |
> | ----------- | ------------------ | --------------- | ----------------- |
> | **5**       | −0.6 / +0.6        | +1.0 / +1.5     | −0.6 / −0.4       |
> | **15**      | −0.2 / −0.2        | +0.8 / +0.3     | −0.2 / +1.7       |
>
> *(All values represent percentage-point differences relative to bonus = 10.)*
>
> The results indicate only **minor fluctuations**  across datasets and metrics ( substantially smaller than the performance gaps observed in Table 2) with no consistent trend favoring smaller or larger bonus values. This confirms that:
>
> > **Model performance is insensitive to the specific bonus magnitude as long as it remains a higher-order reward.**
>
> Thus, both the design and the empirical behavior of the bonus reward demonstrate that our reward shaping is principled, stable, and tightly aligned with the learning dynamics of structured RAG reasoning—rather than heuristic or arbitrary.

---

> > ### Author Response · Authors · 2025-11-19
> >
> > ---
> >
> > **W2, W3 — On conceptual clarity around transparency/robustness and technical novelty**
> >
> > Recent RL-based RAG systems primarily focus on **dynamic retrieval** (e.g., multi-step search, web queries) like Search-R1, R1-Searcher, DeepResearcher, where the key to improving reasoning lies in obtaining better supporting paragraphs. In contrast, our work addresses a *different and under-explored challenge*: **how to make the generator itself transparent, auditable, and robust given a fixed set of retrieved evidence**. This is the central conceptual novelty of our framework.
> >
> > In Section 3.1 (*Framework Formalization*), we define a unified, transparent RAG reasoning process consisting of:
> >
> > * **Evidence selection**,
> > * **Structured reasoning trace**,
> > * **Citation–reasoning alignment**.
> >
> > To achieve this, we design **task-aligned reward components** that directly supervise these behaviors through reinforcement learning. However, training such a structured generator is unstable under traditional KL estimation; therefore, we introduce a stability requirement and adopt the k2 estimator.
> >
> > The theoretical advantages of k2 over k3 have been established by prior work [1]. Our contribution is **empirically demonstrating**, for the first time, that k2 leads to more stable training and better structured reasoning in RAG generators. Across:
> >
> > * comparative experiments (Table 3),
> > * out-of-domain tests (Figure 2, Table 4),
> > * ablation studies (Figure 3),
> > * analytical experiments (Appendix H, K),
> > * case studies (Figure 4),
> >
> > we consistently find that **our improvements do not arise from the output template**, but from **transparency-oriented rewards + stable KL-regularized training**, which meaningfully enhance multi-hop reasoning quality.
> >
> > Together, these components serve a unified and novel research objective: **transparent and stable RAG generation**, providing a practical and effective direction for real-world RAG deployment.
> >
> > ---
> >
> > **W4 — On evaluation scope and complex benchmarks**
> >
> > Browsecomp, simpleQA, and GPQA indeed involve more complex reasoning pipelines, but they also rely on **advanced retrievers, multi-stage systems, or lack evidence annotations entirely**. Because our contribution is orthogonal to retrieval complexity—we aim to improve **generation transparency given the same retrieved evidence**—these datasets are not well suited for isolating and evaluating our method.
> >
> > To demonstrate generality beyond a single dataset, we include **out-of-domain generalization experiments** (Figure 2, Table 4), showing that our method transfers well across datasets with differing structure and difficulty.
> >
> > We will incorporate this clarification in the revised version and expand the discussion on broader evaluation contexts and potential future integration with stronger retrievers.
> >
> > ---
> >
> > **Q1 — On the missing reference**
> >
> > The cited work [1] on KL estimation is already included in our original submission, in Section 2.2, line 107. This line of research inspired our investigation of KL estimator choice for transparent RAG generation.
> >
> > ---
> >
> > **Q2 — On general KL estimator selection**
> >
> > We agree that selecting KL estimators for **general-purpose** RL is an important research question. However, our focus in this paper is: **how KL estimation affects the stability and transparency of a structured RAG generator**. We deliberately scope the discussion and experiments to this setting, and view general KL estimator selection as valuable future work.
> >
> >
> > ---
> >
> > [1] On a few pitfalls in KL divergence gradient estimation for RL

---

### Official Review · Reviewer_JpRb · 2025-10-31

**Soundness:** 2
**Presentation:** 3
**Contribution:** 2
**Rating:** 4
**Confidence:** 4

**Summary:**

The paper proposes ARENA, an RL-trained RAG framework that aims to improve (i) transparency via a structured output protocol with explicit citations and (ii) stability via a stabilized KL term by using the k2 estimator. An adaptive reward combines format, answer correctness, citation relevance, and a large “all-correct” bonus. Experiments on HotpotQA, 2Wiki, and MuSiQue report gains over several baselines, with additional generalization and ablation studies. The work contrasts k2 vs. the commonly used k3 estimator for the KL regularizer in GRPO, arguing for lower variance and better stability for k2, with a brief analysis around gradients of D_KL.

**Strengths:**

1. This paper presents a clear, reproducible, and structured protocol that is easy to use and auto-grade.
2. This paper has broad empirical coverage (multiple datasets, retrieval settings, and model sizes) and some ablations.
3. This paper has some practical engineering contributions: reward design and training configuration are described in enough detail to be reimplemented.

**Weaknesses:**

1. The paper lists "transparency" and "stability" as two major challenges, and the corresponding solutions (structured protocols/reward and KL estimator replacement) are almost entirely independent of each other. It looks like two different ideas are stacked into the same paper.

2. Table 4: This table only compares different "training combinations" for ARENA, which has limited significance for intrinsic comparison. It lacks a comparison with strong baselines under the same training combination settings. Without this, it is unclear whether the improvements come from data mixing or the method itself.

3. The k2 estimator used in the paper is maybe similar to approaches in existing works (e.g., "Rethinking KL Regularization in RLHF: From Value Estimation to Gradient Optimization"). It seems more like adopting an off-the-shelf alternative rather than introducing a novel theoretical method. The paper does not clearly articulate the fundamental differences or incremental contributions compared to these prior works.

4. k2 appears more like a switchable implementation or hyperparameter choice rather than a core methodological innovation. The paper lacks comprehensive comparisons and statistical significance tests for robustness.

5. Figure 5: The axis scales are severely inconsistent (left: 0.0–0.5; right: 0–5,000,000), making the comparison misleading. The scales and units should be unified, or the reason for the differing dimensions must be explained.

6. Insufficient Evaluation of "Transparency/Explainability": The authors mainly rely on case studies and custom metrics to claim improvements, but there is a lack of systematic quantification of citation accuracy, coverage, and redundancy, as well as fair comparisons with other methods. The current "relevance score" is strongly coupled with the training reward, which can lead to overestimation.

7. Ambiguity in Prompt-based Setup: The so-called "answering questions directly using references" does not clarify how the references are obtained, whether the methods share the same retrieval results, or whether there is any information leakage. This affects the fairness of comparisons.

8. Incomplete Related Work: The paper lacks a systematic discussion and empirical comparisons with relevant works in the "attributable citation" RAG direction, such as[1][2]

[1] Citation-Enhanced Generation for LLM-based Chatbots (ACL 2024)

[2] Model Internals-based Answer Attribution for Trustworthy RAG (EMNLP 2024)

**Questions:**

Please address the questions in the Weaknesses.

---

> ### Author Response · Authors · 2025-11-19
>
> Thank you for recognizing the clarity and reproducibility of our structured protocol, the breadth of our empirical evaluation, and the practical value of our implementation details. Our goal is to explore how transparency-oriented reinforcement learning can enable RAG generators to produce stable, evidence-grounded, and traceable reasoning paths. Below, we respond to each concern in detail.
>
> ---
>
> **W1 — Relationship between transparency and stability**
>
> We believe that a strong RAG generator should **stably produce high-traceability outputs**, where both the reasoning process and evidence usage are explicit and verifiable. To achieve this, we introduce a **structured output protocol** and train the model through reinforcement learning to internalize this reasoning behavior. However, during RL training, traditional KL estimators often cause instability, leading to collapsed or malformed structures. Therefore, we investigate more stable KL estimation strategies to ensure consistent learning of structured reasoning.
>
> Under this unified perspective, **our transparency and stability solutions serve the same overarching goal**: producing reliable, structured, and evidence-traceable RAG outputs. Transparency defines the structure we want; stable KL ensures the model can maintain it throughout training.
>
> ---
>
> **W2, W7 — On fairness of comparisons and use of retrieved contexts**
>
> Our work focuses on **how to generate more reliable reasoning paths given the same retrieved evidence**, not on improving retrieval itself. As stated in Section 4.4, lines 351–353:
>
> > “During inference, we concatenate all retrieved paragraphs provided by the dataset in their original order.”
>
> These retrieved contexts are **human-annotated references** from the datasets. Appendix C documents the full labeling process and preprocessing pipeline. Section 4.3, lines 320–322 also clarifies:
>
> > “For consistency, we use their released checkpoints and replace their retrieved contexts with those provided by datasets.”
>
> Thus, **all methods—including ours and all baselines—are evaluated using exactly the same set of supporting paragraphs**.
>
> In contrast, prior RAG works integrate independent retrieval modules, using either local Wikipedia corpora or web search engines. Because these methods rely on different retrieval pipelines, **it is not feasible to retrain them with identical training inputs**, and indeed these prior works themselves evaluate each baseline using its own retrieved results. Our setup removes this variability and ensures the fairest possible comparison by unifying the retrieved evidence across all models.
>
> ---
>
> **W3, W4 — On the role of the k2 estimator and its empirical contribution**
>
> Prior theoretical research has shown that the **k2 estimator provides a closer approximation of KL gradients** than k3 and enjoys lower variance and non-negativity guarantees. As noted by the reviewer, this family of estimators has been explored before—including the reference [1], which we had already cited in Appendix B, lines 794–795 of our initial submission.
>
> Our contribution is **empirical rather than theoretical**: we demonstrate for the first time that k2 brings significant **practical stability benefits** when training a *structure-dependent RAG generator*. In GRPO, KL is part of the loss and therefore its smoothness directly affects policy updates. Multi-hop reasoning tasks involve sparse, discontinuous, and highly structured reward signals, under which **k3 introduces extreme gradient spikes**, causing abrupt policy shifts, which in turn lead to **malformed or incomplete structured responses**, resulting in rapid deterioration of evidence tracing and reasoning quality.
>
> Empirical evidence supports these observations:
>
> * **Figure 3** shows structural adherence breakdown when using k3.
> * **Figure 5** demonstrates that k3 produces extreme KL spikes (right), while **k2 yields smooth, stable convergence** (left).
>
> Thus, our work provides the **first empirical link** between KL-estimator stability and transparent multi-hop RAG reasoning performance, extending prior theoretical insights into a new application domain.
>
> ---
>
> **W5 — On axis scaling in Figure 5**
>
> Figure 5 aims to illustrate a crucial qualitative difference: **using k3 leads to extreme, out-of-range KL spikes**, whereas k2 remains within a normal and stable range. Using identical axis scales would either (a) collapse the k2 curve into an indistinguishable flat line, or (b) push the k3 spike far beyond the visible region. Either approach would obscure the intended comparison. We therefore chose axis scales that best communicate the contrasting behaviors, while preserving accurate values. We will clarify this rationale in the revised paper.

---

> > ### Author Response · Authors · 2025-11-19
> >
> > ---
> >
> > **W6, W8 — On evaluating transparency/explainability and related work**
> >
> > In the Introduction and Related Work sections, we already discuss advances in the “attributable citation” direction; the work [2] is already cited in Introduction, lines 39-40, and our Related Work section already includes several works similar to [3] (like [4,5]), we will incorporate [3] along with additional relevant studies in the revised version. Existing methods mainly add **post-hoc citation annotations** to generated sentences but do not evaluate whether the model selects *correct* supporting paragraphs (due to small top-k retrieval sizes, often ≤5), nor do they employ modern RL-based training strategies to improve evidence attribution.
> >
> > As modern LLMs can process **larger retrieved contexts (≥10 paragraphs)**, a key challenge becomes evaluating whether the model can **identify relevant evidence from a larger candidate pool**. For this reason, we design the **relevance metric** to directly measure citation accuracy under larger context sizes.
> >
> > We demonstrate the effectiveness of our method through:
> >
> > * comparative experiments (Table 3),
> > * out-of-domain tests (Figure 2),
> > * ablation studies (Figure 3),
> > * analytical experiments (Appendix K),
> > * case studies (Figure 4).
> >
> > We attempted to evaluate other RL-based models’ citation ability using our structured protocol, but because their training did **not** include relevance-labeled prompts, these models **could not maintain structured output format**, making direct comparison infeasible.
> >
> > Taken together, our relevance metric and experimental results provide a **systematic and evidence-based evaluation** of transparency/explainability in RAG generation.
> >
> > ---
> >
> > [1] Rethinking KL Regularization in RLHF: From Value Estimation to Gradient Optimization
> >
> > [2] Citation-Enhanced Generation for LLM-based Chatbots
> >
> > [3] Model Internals-based Answer Attribution for Trustworthy RAG
> >
> > [4] Citation: A key to building responsible and accountable large language models
> >
> > [5] Bridging relevance and reasoning: Rationale distillation in retrieval-augmented generation

---

### Official Review · Reviewer_rT9q · 2025-10-31

**Soundness:** 2
**Presentation:** 2
**Contribution:** 2
**Rating:** 4
**Confidence:** 4

**Summary:**

This paper proposes an Adaptive-Reward Reinforcement Learning (AR-RL) framework for improving the transparency and robustness of Retrieval-Augmented Generation (RAG) models. The framework aims to make the RAG process more interpretable and stable by introducing three components:
(1) a Stability KL term to ensure training stability,
(2) a Transparency Module that encourages explicit citation behavior, and
(3) a reward design that evaluates response relevance and citation correctness.
The goal is to produce RAG outputs that are both faithful to retrieved evidence and explainable through citation traces. Experiments on QA and citation-grounded datasets show that the proposed approach improves citation precision and factuality metrics over baseline RAG and reinforcement learning variants.

**Strengths:**

1. tackling the issue of traceable and interpretable RAG outputs is meaningful and timely.
2. includes modules addressing both stability and traceability, leading to more robust empirical performance.
3. results are consistently presented and compared with competitive baselines.
4. transparency-driven reward optimization aligns with current research trends in trustworthy RAG and factual QA.

**Weaknesses:**

1.	Weak conceptual connection among modules — The three main components (Stability KL, Transparency module, and Adaptive Reward) do not form a tightly integrated theoretical framework. Stability is for smoother training, transparency is for citation generation, and reward shaping is for content quality; however, the paper does not clearly articulate how these together solve one unified problem. This makes the work appear as a collection of engineering heuristics rather than a cohesive research contribution.
	2.	Reward design vulnerability — The Relevance Reward assigns 0.5 for partial overlaps and 0 for no overlaps, which may incentivize reward hacking: the model can increase citation quantity to maximize overlap probability, reducing citation precision. Introducing a citation-length penalty or constraint reward could prevent this issue.
	3.	Potential bias in ground-truth construction — The relevance labels come from locating paragraphs containing annotated supporting facts. This process assumes that only those paragraphs are relevant, but multiple documents might contain semantically similar information. As a result, the reward may penalize correct but alternative citations, introducing redundancy and bias. A better approach would be sentence-level citation matching and reference-grouping, where one correct reference per knowledge group yields a high reward.
	4.	Limited evaluation scope — The experiments are restricted to simpler QA datasets such as HotpotQA. Modern reasoning-oriented RAG tasks like BC and GAIA involve more complex multi-hop and abductive reasoning. Evaluating on such benchmarks would better demonstrate the framework’s robustness and generality.
	5.	Moderate novelty — Most components (stability KL, adaptive reward, transparency regularization) are known techniques; the innovation mainly lies in combining them, not in introducing new theoretical insights.

**Questions:**

1.	How are the three components—Stability KL, Transparency Module, and Adaptive Reward—conceptually linked? Could the authors provide a unified view or optimization objective that justifies their joint inclusion?
2.	Have you observed any evidence of reward hacking (e.g., the model generating excessive citations to gain partial-overlap rewards)? If so, how is it mitigated?
3.	Can you clarify how the ground-truth relevance labels are constructed when multiple supporting documents contain similar information?
4.	How would the framework perform on reasoning-heavy RAG benchmarks (e.g., BC, GAIA) where relevance and correctness may not perfectly align?
5.	Would adding a citation-length penalty or dynamic normalization improve reward fairness and prevent over-citation behaviors?

---

> ### Author Response · Authors · 2025-11-19
>
> We thank the reviewer for the constructive feedback and for recognizing the importance of transparent and stable RAG generation. Our goal is to explore how transparency-oriented reinforcement learning can improve evidence tracing and multi-hop reasoning quality in RAG generators. We address all concerns point-by-point below.
>
> ---
>
> **W1, W5, Q1 — On unified objective and conceptual linkage among components**
>
> The unified research question we aim to address is: **How to obtain a high-quality, traceable RAG generator whose reasoning chain is explicit, verifiable, and robust?**
> All three components in our framework directly contribute to this shared objective:
>
> * **Transparency Module** provides *verifiable and auditable citation chains*, ensuring that each reasoning step is grounded in evidence that can be checked and traced.
> * **Adaptive Rewards** supply explicit *supervision for transparent reasoning*, combining format, accuracy, and relevance signals into a single multi-objective target aligned with traceable RAG generation.
> * **Stable KL (k2 estimator)** ensures *smooth and reliable parameter updates*, preventing the model from collapsing into malformed or incomplete structures and enabling consistent production of the structured protocol required for traceability.
>
> Together, these components form a coherent optimization framework aimed at **stable, evidence-grounded, and interpretable RAG generation**, rather than a collection of independent heuristics.
>
> ---
>
> **W2, Q2, Q5 — On reward hacking, citation length, and normalization**
>
> To analyze whether the relevance reward (0.5 for partial overlaps) causes citation-length inflation or reward hacking, we report the following statistics across datasets:
>
> | Dataset | Avg # Evidence IDs | GT # Supporting IDs | Relevance Score (%) |
> | -------- | ----------------- | ------------ | --------------- |
> | **HotpotQA** | 2.01              | 2.00            | 91.3            |
> | **2Wiki**    | 2.29              | 2.19         | 90.8            |
> | **MuSiQue**  | 2.06              | 2.73         | 59.5            |
>
> The **citation-length remains stable and close to ground-truth** on all datasets, showing no indication of the model artificially increasing the number of cited paragraphs to exploit partial-overlap rewards. This stability arises because our **bonus reward assigns a significantly larger value only when the model achieves full correctness (format + accuracy + relevance)**. As a result, the model is strongly incentivized to pursue *complete* and *precise* grounding rather than staying in partially overlapped states.
>
> While citation-length penalties or dynamic normalization could be incorporated in future extensions, our current results indicate that the model already avoids over-citation and behaves consistently across testsets.
>
> ---
>
> **W3 and Q3 — On ground-truth relevance labels and overlapping information**
>
> Our relevance labels are derived from the **reference lists provided by each dataset**, which originate from their human-annotated supporting facts. These datasets deliberately avoid assigning multiple redundant documents containing identical information to the same question; therefore, during training, there is **no scenario where the model selects a paragraph containing correct information but receives no credit**, aligning well with our training objective.
>
> In real retrieval settings, it is true that multiple documents may contain semantically similar evidence. However, in such cases there is **no single gold reference list**, and relevance labels therefore serve the purpose of *transparency and auditability*, not correctness evaluation. Any document containing the correct information would still provide a transparent and auditable reasoning trace. The **relevance scores reported in our paper serve precisely this role**, indicating how faithfully the model grounds its reasoning in retrievable evidence.
>
> We agree that future work would benefit from benchmarks that explicitly include redundant or semantically overlapping evidence sources, and we hope the community will move toward such testsets.
>
> ---
>
> **W4 and Q4 — On scope of evaluation and complex RAG benchmarks**
>
> BC and GAIA are indeed more complex benchmarks involving multi-step inductive reasoning and typically require **advanced retrievers and multi-stage pipelines**. Our contribution is orthogonal to retrieval complexity: we focus on **improving the transparency and effectiveness of the generation process given the same retrieved evidence**, rather than improving retrieval itself. Therefore, these tasks are not ideally suited to isolating and evaluating our contribution.
>
> We will incorporate this clarification in the revised version and plan to discuss how our framework may be combined with sophisticated retrievers in future work.

---

> > ### Comment · Reviewer_rT9q · 2025-11-27
> >
> > The authors have addressed most of my concerns in their rebuttal; therefore, I will revise my scores accordingly.

---

> > > ### Author Response · Authors · 2025-11-28
> > >
> > > We truly appreciate your constructive feedback and are glad our clarifications were helpful.

---

### Official Review · Reviewer_BLQC · 2025-10-31

**Soundness:** 3
**Presentation:** 3
**Contribution:** 3
**Rating:** 6
**Confidence:** 3

**Summary:**

This paper proposes ARENA, a transparent and robust RAG generator framework trained via RL with adaptive rewards. While ARENA outperforms naive GRPO and other popular methods with better RAG transparency, its overall novelty is limited due to the reliance on existing technical components (e.g., k2 KL estimator, multi-objective linear reward).

**Strengths:**

S1：ARENA significantly improves the performance of RAG generators compared to naive GRPO and other baseline methods (prompt-based, SFT-based, RL-based) across three multi-hop QA datasets (HotpotQA, 2Wiki, MuSiQue). It successfully balances transparency (explicit evidence-reasoning-answer traces) and stability (mitigated gradient spikes), providing actionable insights for RL-based RAG research.

S2：The introduction of a bonus reward effectively incentivizes the model to simultaneously satisfy format adherence, answer accuracy, and evidence relevance throughout training. Ablation studies confirm that this component is critical for maintaining consistent structured outputs, addressing a common challenge in multi-objective RL for language generation.

S3：The structured output protocol (with <relevance>, <analysis>, and <answer> sections) enhances decision traceability, a key requirement for real-world RAG deployment. Quantitative evaluations (Format/Relevance scores in Table 3) demonstrate that ARENA achieves near-perfect format adherence and substantially higher citation accuracy than base models.

**Weaknesses:**

W1：The "Adaptive Reward" component, while task-specific, relies on a simple linear combination of four reward terms (format, accuracy, relevance, bonus) with equal weights. This multi-objective reward design is not novel. The paper provides no justification for omitting dynamic weight adjustment or more advanced reward shaping strategies.

W2：While the paper adopts the k2 KL estimator in the GRPO pipeline to stabilize training, the core advantages of k2 (non-negativity, lower variance, gradient equivalence) are not original to this work (as acknowledged in Section 3.2).

W3：Although the paper includes theoretical analysis of the k2 estimator (Appendix B), it lacks in-depth discussion of why gradient stability (achieved via k2) translates to better RAG performance. The link between smoother KL curves and improved evidence selection/reasoning quality in multi-hop QA remains under-explained.

W4：Further experiments (e.g., generalization, cross-task transfer) only report results for ARENA-Qwen2.5-7B, neglecting ARENA-Llama3.1-8B and limiting insights into cross-architecture generalization.

**Questions:**

Q1：For the results reported in Table 2: Are all RL-based and SFT-based baselines trained on the Qwen2.5-7B backbone? If so, does the lack of Llama3.1-8B-based baselines make the performance comparison between ARENA-Llama3.1-8B and other models unfair (due to backbone differences)?

Q2：The paper claims that k2 estimator improves training stability, but it does not explain how this stability specifically benefits multi-hop reasoning and evidence tracing. Could the authors elaborate on the mechanism linking gradient smoothness to better RAG performance?

Q3：The bonus reward (10 points) is significantly larger than the other three component rewards (1 point each), yet the paper provides no justification for selecting the value "10" or conducting sensitivity analysis on this hyperparameter. Why was the specific value of 10 chosen, and have experiments been conducted with alternative bonus values (e.g., 5, 15) to validate its optimality for RAG multi-hop QA tasks?

---

> ### Author Response · Authors · 2025-11-19
>
> We thank the reviewer for the thoughtful and constructive comments. We appreciate the positive recognition of ARENA’s improvements in transparency, stability, and multi-hop QA performance. Below we address each concern in detail and clarify the novelty of our contributions.
>
> ---
>
> **W1 – On the simplicity of the linear adaptive reward**
>
> Recent RAG-oriented RL works following *DeepSeek-R1* (e.g., Search-R1, R1-Searcher, ZeroSearch) consistently demonstrated that **rule-based linear reward combinations** are *effective* for activating reasoning abilities in RL-based LLMs. They typically focusing on *retrieval quality*, while our work emphasizes a different research objective:
>
> > **“Adding an explicit transparency-oriented, RAG-specific reward that encourages accurate citation and evidence grounding significantly improves multi-hop reasoning and decision traceability.”**
>
> rather than
>
> > **“Designing dynamic reward weighting schedules during training.”**
>
> We acknowledge that several very recent studies explore **dynamically adjusting reward weights** based on rollout statistics. This is an interesting direction, and we plan to investigate such strategies for RAG-centric RL in future work.
>
> ---
>
> **W2, W3, Q2 – On k2 KL estimator stability and its effect on multi-hop RAG**
>
> Prior theoretical work has rigorously shown that the **k2 estimator more faithfully approximates KL divergence gradients** than k3, with lower variance and guaranteed non-negativity. Our contribution is **empirically demonstrating its practical benefits for RAG-specific RL**.
>
> In GRPO, **KL is part of the training loss**, and therefore its smoothness directly influences policy updates. Multi-hop QA presents discontinuous, sparse, and highly structured reward signals. Under such conditions:
>
> * **k3 produces gradient spikes**, which lead to abrupt policy shifts.
> * These shifts cause the model to frequently output **malformed or incomplete structured responses**, violating the required `<relevance>–<analysis>–<answer>` protocol.
> * As a consequence, **evidence tracing and reasoning quality degrade sharply**, because a single formatting error collapses the entire reward.
>
> Empirically (Figure 3), we observe that with k3 the model progressively loses structural adherence during training, whereas **k2 maintains smooth KL curves and stable structural formatting**, enabling consistent evidence grounding and multi-hop reasoning.
>
> Thus, our work provides the *first empirical evidence* linking KL estimator stability to **transparent, structure-dependent RAG reasoning quality**, which extends prior theoretical findings.
>
> ---
>
> **W4, Q1 – On baseline fairness and cross-architecture generalization**
>
> All 7B-scale RL-based and SFT-based baselines compared in Table 2 were trained on **Qwen2.5-7B or Qwen2.5-7B-Instruct**, ensuring a **fair, backbone-consistent comparison** with ARENA-7B.
>
> We agree with the reviewer that this point should be clarified more explicitly. In the revised version, we will highlight that:
>
> * ARENA-Llama3.1-8B is shown only as a cross-architecture demonstration, not directly compared with Qwen-based baselines.
> * All fairness-sensitive comparisons use matching backbone models.
>
> Our main experiments already show that the ARENA training pipeline transfers well across different model architectures, and subsequent analytical experiments further validate the effectiveness and generality of our approach.

---

> > ### Author Response · Authors · 2025-11-19
> >
> > ---
> >
> > **Q3 – On the choice and sensitivity of the bonus reward value**
> >
> > Our ablation study (Appendix I) shows that removing the bonus reward causes the model’s **format adherence to deteriorate during training**. We believe this happens because with multiple reward components, many “partially correct” outputs receive identical total rewards, making it difficult for GRPO’s group-based advantage to prioritize fully correct answers.
> >
> > To address this, we introduce a **higher-magnitude bonus reward** that encourages the model to learn the rare but essential event of being *simultaneously* correct on format, accuracy, and relevance. The choice of **10** is not the result of hyperparameter tuning, but rather a **safe scale separation** that makes the bonus significantly larger than individual components (1 point each), while preserving optimization stability.
> >
> > Below we present a concise comparison showing the **performance deltas** of bonus values 5 and 15 relative to the default **bonus = 10**. Positive numbers indicate better performance than bonus = 10; negative numbers indicate lower performance.
> >
> > | Bonus Value | HotpotQA ΔEM / ΔF1 | 2Wiki ΔEM / ΔF1 | MuSiQue ΔEM / ΔF1 |
> > | ----------- | ------------------ | --------------- | ----------------- |
> > | **5**       | −0.6 / +0.6        | +1.0 / +1.5     | −0.6 / −0.4       |
> > | **15**      | −0.2 / −0.2        | +0.8 / +0.3     | −0.2 / +1.7       |
> >
> > *(All values represent percentage-point differences relative to bonus = 10.)*
> >
> > Although small fluctuations exist, they are substantially smaller than the performance gaps observed in Table 2, and no consistent improvement trend favors either smaller or larger bonus values. This confirms that the model’s performance is robust to the precise choice of the bonus, as long as the bonus maintains its role as a higher-order reward.

---

### Note · Authors · 2026-01-16

I have read and agree with the venue's withdrawal policy on behalf of myself and my co-authors.